# Impact of Forage Sources on Ruminal Bacteriome and Carcass Traits in Hanwoo Steers During the Late Fattening Stages

**DOI:** 10.3390/microorganisms12102082

**Published:** 2024-10-17

**Authors:** Ryukseok Kang, Jaeyong Song, Joong Kook Park, Sukjun Yun, Jeong Heon Lee, Jun Sang Ahn, Chaemin Yu, Geonwoo Kim, Jongsik Jeong, Myeong-Gwan Oh, Wanho Jo, Woohyung Lee, Mekonnen Tilahun, Tansol Park

**Affiliations:** 1Department of Animal Science and Technology, Chung-Ang University, Anseong 17546, Republic of Korea; sixstone1@cau.ac.kr (R.K.); qjsro2186@naver.com (G.K.); omg0525@cau.ac.kr (M.-G.O.); dmtilahun84@gmail.com (M.T.); 2Nonghyup Feed Co., Ltd., Seoul 05398, Republic of Korea

**Keywords:** Hanwoo steers, forage source, ruminal bacteriome, alpha-diversity, meat yield

## Abstract

This study examined the effects of different forage sources on the ruminal bacteriome, growth performance, and carcass characteristics of Hanwoo steers during the fattening stage. In Korea, where high-concentrate feeding is common, selecting suitable forage is crucial for sustainable beef production. Fifteen 23-month-old Hanwoo steers, weighing an average of 679.27 ± 43.60 kg, were fed the following five different forage sources: oat hay (OAT), rye silage (RYE), Italian ryegrass (IRS), barley forage (BAR), and rice straw silage (RSS), alongside 1.5 kg of dry matter concentrate daily for five months. Carcass traits were evaluated post-slaughter, and rumen fluid samples were analyzed using full-length 16S rRNA gene sequencing to determine the bacteriome composition. The forage source significantly affected the alpha-diversity indices and bacteriome biomarkers linked to the feed efficiency and ruminal fermentation. Differences in the backfat thickness and meat yield index were noted, with alpha-diversity indices correlating with carcass traits. The phylum Planctomycetota, especially the family Thermoguttaceae, was linked to nitrogen fixation in high-protein diets like IRS, while the genus *Limimorpha* emerged as a biomarker for the meat yield. These findings highlight the importance of forage selection during late fattening to optimize beef production, considering diet and bacteriome shifts.

## 1. Introduction

Up to 90% of the diet used to produce highly marbled meat by the beef cattle industry in Korea during the middle and late fattening stages was high-concentrate feed. Despite the relatively small proportion of forage in beef cattle diets, selecting appropriate forages during the fattening stage is important for economic and sustainable beef production [1,2].

Feeding cattle with locally available high-quality forage can reduce feed transportation costs [3]. The forage source affects the beef cattle carcass characteristics. Previous studies have analyzed the effect of feeding beef cattle with different forage sources on the growth performance, carcass weight, fat deposition, and meat quality. In South Korea, different forage sources are used to feed beef cattle when considering the different regions; however, feeding cattle with bagged rice straw silage (RSS) is the most common. However, high-quality locally produced forages, such as Italian ryegrass (IRS), rye silage (RYE), and green barley forage (BAR), are also widely used. Furthermore, the diversity of locally produced forage items is increasing in South Korea. A survey conducted in 2021 on the application of domestic forage in Korean beef farms revealed that 43.5% of farms fed beef cattle with rice straw, 19.4% fed beef cattle with IRS, and 13.2% with fed beef cattle with RYE and BAR [4].

Several locally grown forages have been used to feed beef cattle. IRS has been recommended as a pasture or forage for total mixed rations in beef cattle feed [5] for increased daily weight gain, feed efficiency, backfat thickness, and rib eye area [6]. A study reported that feeding cattle with RYE alone or combined with other silages did not significantly affect the live weight gain, carcass gain, carcass characteristics, meat quality, or fatty acid composition [7]. Feeding post-weaning calves with oat hay (OAT) increased the crude protein digestibility, ruminal fermentation, and nitrogen utilization, and reduced diarrhea [8]. Feeding with whole-crop BAR significantly increased the growth performance, meat quality, feed intake, carcass traits, meat price, palatability, and fat content of Hanwoo steers [9] compared with those of rice straw, which is the primary forage used in Korea, Therefore, feeding cattle with alternative forages improves the overall beef cattle performance without additional costs.

Various forages can influence the growth performance, carcass weight, fat deposition, and meat quality attributes, highlighting the importance of selecting the appropriate forage to achieve the desired carcass traits. However, studies comparing domestic forage sources during the late fattening stages in Korea are limited.

Ruminal microorganisms, primarily bacterial populations, play a crucial role in ruminant digestion [10]. A previous study using Hanwoo steers noticed the differential appearance of the ruminal bacteriota composition on two animal groups with extreme meat quality indices [11]. Two studies using different breeds of beef steers suggested that low richness indices of ruminal bacteria may correlate with high marbled meat production [11,12]. Biomarkers associated with feed efficiency and average daily gain, such as butyrate- and propionate-producing bacteria and specific taxa within the Proteobacteria, have been identified [13]. However, further research is required to confirm the reliability of these biomarkers.

Therefore, we hypothesized that feeding Hanwoo beef steers with different forages during the fattening stages influences the growth performance and carcass characteristics by changing the ruminal bacteriome. Furthermore, this study investigated the correlation between the ruminal bacteriome and animal performance by analyzing full-length 16S rRNA gene sequences using predicted functional features.

## 2. Materials and Methods

### 2.1. Animal Feeding Trial and Rumen Fluid Sampling

This experiment was conducted at the Nonghyup Research Farm, Republic of Korea, considering the Hanwoo Care and User Guidelines established by the Institutional Animal Care and Use Committee of Chung-Ang University (202401030036). Fifteen 23-month-old Hanwoo steers, with an average body weight of 679.27 ± 43.60 kg, were randomly assigned to individual pens. The cattle were fed the experimental diets twice daily at 09:00 h and 16:00 h. Water and mineral blocks were fed ad libitum. The nutritional compositions of the experimental diets are presented in Table 1. The animals were fed with oat hay (OAT), rye silage (RYE), Italian ryegrass (IRS), green barley forage (BAR), and rice straw silage (RSS) at 1.5 kg dry matter and 9.0 kg dry matter of concentrate per day. Each of the dietary treatments were fed to the assigned animals for 5 months. The dry matter intake (DMI) was measured for three consecutive days each month during the experiment, and the DMI was determined by measuring the remaining feed before the morning feeding. The average daily gain was determined by recording body weight at 9:00 AM each month. Upon completion of the feeding trial, approximately 200 mL of rumen fluid was collected from each Hanwoo steer by stomach tubing 2 h post-feeding. The rumen fluid samples were then transferred to the laboratory in 39 °C preheated thermal bottles and subjected to metagenomic DNA extraction and subsequent bacteriome analysis. At the end of the experimental period, all of the animals were slaughtered at the National Agricultural Cooperative Federation, Republic of Korea, a commercial abattoir.

### 2.2. Chemical Analysis of Experimental Feeds

Random grab samples of the OAT, RYE, IRS, BAR, and RSS were ground and sieved through a 1 mm screen for proximal analysis. The OAT dry matter content (DM) was determined after drying the samples using an air-dryer (Thermo Fisher Scientific, Waltham, MA, USA) at 65 °C for 3 days, following AOAC methods [13]. The forage silages were dried in a freeze-dryer (Thermo Fisher Scientific) at −80 °C for 3 to 5 days to determine their DM content. Crude protein, ether extract, crude ash, neutral detergent fiber, and acid detergent fiber were determined according to the methods described previously by Serrapica et al. [14] and the AOAC methods [15].

### 2.3. Analysis of Carcass Characteristics

The meat quantity and quality were evaluated according to the Animal Products Grading Service Manual [16], which evaluates the quantity grade, marbling score, meat color, fat color, texture, maturity, and quality grade. Backfat thickness was measured perpendicularly to the outer surface at two-thirds of the length of the rib eye, between the last rib and the first lumbar vertebra. The rib eye area on the cut surface was measured using a standard grid. The meat yield index was calculated based on the Korean carcass grading procedure [17] through the following equation:Meat yield index = 68.184 − [0.625 × backfat thickness (mm)] + [0.130 × rib-eye area (cm^2^)] − [0.024 × carcass weight (kg)] + 3.23

The meat yield index ranged from 60.05 to 65.72, resulting in quantity grades from grade C to B. Marbling was graded from 1 (low fat) to 9 (high fat). The meat color was evaluated on a scale from 1 (very bright cherry red) to 7 (very dark red), and the fat color on a scale from 1 (white) to 7 (yellow). The texture was evaluated on a scale from 1 (very smooth) to 3 (very coarse). Maturity was assessed on a scale from 1 (1 to 15 mm, young) to 2 (15 to 26 mm, old). The quality grades were evaluated based on the marbling score, resulting in grades 1+ to 1++ in this study.

### 2.4. Analysis of Ruminal Bacteriome

The total metagenomic DNA was extracted from 15 ruminal fluid samples using the repeated bead-beating and column purification method described by Yu and Morrison [18]. To amplify the full-length 16S rRNA gene amplicons, a library was generated from each DNA sample using the primers 27F (5′-AGRGTTYGATYMTGGCTCAG-3′) and 1492R (5′-GYTACCTTGTTACGACTT-3′) [19], and sequenced at Macrogen Inc., Seoul, Republic of Korea, using a PacBio Sequel IIe system in long-read HiFi mode (Pacific Biosciences, Menlo Park, CA, USA). The resulting sequences were analyzed using QIIME2 (version amplicon 2024.02) [20] as described in a previous study [21]. Briefly, the primers were trimmed using Cutadapt (version 4.6) [22], followed by quality filtering (q ≥ 25), denoising, and the removal of chimeric sequences using the DADA2 denoise-ccs plugin [23]. Representative sequences were classified using Scikit-learn using the weighted Greengene2 reference database (version 2022.10) [24]. Further taxonomic filtration was performed to remove amplicon sequence variants (ASVs) labeled as “unassigned”, “chloroplast”, or “mitochondria”. An average rarefied abundance table was created by averaging 1000 repeated rarefaction outputs at a specified number of ASVs using q2-repeat-rarefy [25]. Alpha-diversity measurements, including species richness (observed ASVs [26] and Chao1 estimates [27]), evenness [28], Faith’s phylogenetic diversity [29], and Shannon’s [30] and Simpson’s indices [31], were calculated based on the repeatedly rarefied ASV abundance table. The overall ruminal bacteriota between the different forage source-fed groups were compared using principal coordinate analysis based on Bray–Curtis and Jaccard dissimilarity. The functional features of the bacteria predicted from the 16S ASVs were analyzed using the Phylogenetic Investigation of Communities by Reconstruction of Unobserved States 2 [32]. Normalized counts of the predicted enzyme commission (EC) numbers were used to assess the functional dissimilarities between different forage source-fed bacteriomes. The overall distribution of functional profiles was analyzed using Bray–Curtis and Jaccard dissimilarities.

### 2.5. Statistical Analyses

Statistical analyses were conducted on various animal performance data, including live body weights, average daily gain, carcass characteristics, and alpha-diversity measurements. All statistical analyses were conducted using SAS software (version 9.4, SAS Institute Inc., Cary, NC, USA). The significance level was set at α = 0.05 for all tests. Before conducting the primary analyses, the normality of the residuals was assessed using the Shapiro–Wilk test, and the homogeneity of the variance was assessed using Levene’s test. Variables that met the assumptions of normality and homogeneity of variances were analyzed using parametric methods, while variables that violated these assumptions were analyzed using non-parametric tests. For variables that met normality and homogeneity assumptions, a one-way ANOVA was conducted using PROC MIXED in SAS 9.4. The model included treatment as a fixed effect and replication as a random effect, with degrees of freedom calculated using the Kenward–Roger method. The model was specified as follows:Yij = μ + Ti + Rj + εij
where Yij is the response variable, μ is the overall mean, Ti is the fixed effect of the ith treatment, Rj is the random effect of the jth replication, and εij is the random error. Least squares means were calculated for each treatment. Pairwise comparisons between treatments were performed using the Tukey–Kramer adjustment for multiple comparisons. For variables that did not meet normality assumptions, a non-parametric analysis was carried out using the Kruskal–Wallis test (PROC NPAR1WAY with the WILCOXON option). Exact *p*-values were computed to compare these variables across treatments. Results are presented as least squares means ± standard error of the mean (SEM). Differences were considered significant at *p* ≤ 0.05, and trends were discussed at 0.05 < *p* ≤ 0.10. 

Permutational multivariate analysis of variance was used to analyze the overall bacteriota and their functional dissimilarities based on the forage source using the vegan package (version 2.6) [33] and the pairwiseAdonis package (version 0.4.1) [34] in R (4.2.2) with 9999 permutations, followed by multiple-test corrections using the Benjamini–Hochberg method [35]. For bacteriome data, major features, including classified taxa at the phylum, family, genus, and species levels, and predicted functions represented by the ECs, were selected if the average relative abundance of each feature exceeded 0.1% in at least one forage source group. Differentially abundant bacterial phyla, families, genera, and EC numbers were identified using Linear discriminant analysis Effect Size [36] with a *p*-value < 0.05 and a linear discriminant analysis (LDA) score > 2 as the significance threshold. The relative abundances of differential bacterial taxa and predicted EC numbers were correlated with animal performance measurements using Spearman’s correlation analysis via the PROC CORR procedure in SAS 9.4 (SAS Institute, Cary, NC, USA), and the results were visualized using the corrplot package in R [37].

## 3. Results

### 3.1. Animal Production and Carcass Characteristics

The results of the live body weight, average daily gain, and carcass characteristics are presented in Table 2. The effect of different sources of forage (*p* = 0.0412) significantly affected the backfat thickness. Furthermore, a significant pairwise comparison was found between RYE and IRS (adjusted *p* < 0.05). The results of the meat yield index had the same trend as those of the backfat thickness [included in its calculation but with a weak significance (*p* = 0.0992)], with RYE and IRS tending to be different (adjusted *p* < 0.10). Categorical variables related to the meat yield and quality grades exhibited contrasting trends. OAT, RYE, and RSS had a low yield and C grade meat, whereas two-thirds of meat from IRS and BAR fed steers was B grade. All steers fed OAT and RSS had the highest meat quality grade (1++), whereas the other three forage feeding groups had one or two steers with 1+ grade meat. The highest and lowest prices per kilogram of meat were observed in the meat from the OAT- and BAR-fed steers, respectively.

### 3.2. Diversity Measurements of Ruminal Bacteriota

The richness index based on the number of ASVs found in the samples was significantly affected by the forage source (*p* = 0.0299), and the ruminal bacteriota of IRS-fed steers had a significantly higher number of ASVs than those of RYE- and RSS-fed steers (adjusted *p* < 0.05) (Figure 1). The results of Faith’s phylogenetic diversity (*p* = 0.0254) were consistent throughout the evaluated treatments. The ruminal bacteriota of IRS-fed steers had the highest Shannon index, which was significantly higher from that of RYE-fed steers (adjusted *p* < 0.05). With the same discrepancy, a statistical trend was found for Simpson’s index. However, neither the overall bacteriota nor their functional communities were significantly affected by the forage source (*p* > 0.10 [Table 3]).

### 3.3. Identification of Bacterial Biomarkers

Among the major bacterial phyla, families, genera, and species, the dominant taxa associated with each forage source are listed in Table 4. Eubacterium G was the only major taxon dominant in the ruminal bacteriota of OAT-fed steers, whereas Weimeria and its species, Weimeria bifida, were dominant in the ruminal bacteriota of RYE-fed steers. The ruminal bacteriota of IRS-fed steers had six significantly associated taxa, including Planctomycetota, Thermoguttaceae, *Succinivibrio*, *Limimorpha*, *Shuttleworthia*, and *Succinivibrio dextrinosolvens*, while B. Actinobacteria, Coprobacillaceae, and Dethiosulfovibrionaceae were dominant in the ruminal bacteriota of RSS-fed steers. No major taxa were identified as biomarkers in the ruminal bacteriota of BAR-fed steers. The abundances of major unclassified bacterial taxa which were differentially abundant are listed in Appendix A. 

Among the major predicted ECs, seven ECs were dominant in the ruminal bacteriome of RYE-fed steers, such as EC:2.1.1.170 [16S rRNA guanine(527)-N(7)-methyltransferase], EC:2.3.1.30 (serine O-acetyltransferase), EC:2.4.2.10 (orotate phosphoribosyltransferase), EC:2.7.7.60 (2-C-methyl-D-erythritol 4-phosphate cytidylyltransferase), EC:3.6.1.23 (dUTP diphosphatase), EC:4.1.1.20 (diaminopimelate decarboxylase), and EC:5.1.1.3 (glutamate racemase) (Table 5). EC:1.15.1.1 (superoxide dismutase) was the only dominant EC in the ruminal bacteriome of IRS-fed steers. EC:1.1.1.22 (UDP-glucose 6-dehydrogenase) and EC:5.1.1.7 (diaminopimelate epimerase) were dominant in the ruminal bacteriome of BAR-fed steers. EC:2.7.1.71 (shikimate kinase), EC:3.4.11.4 (tripeptide aminopeptidase), and EC:3.5.99.10 (2-iminobutanoate/2-iminopropanoate deaminase) were dominant in the ruminal bacteriome of RSS-fed steers. Among the major predicted MetaCyc pathways, the BAR-dominant RIBOSYN2-PWY [flavin biosynthesis I (bacteria and plant)] was the only pathway associated with the different forages.

### 3.4. Analysis of Significant Correlation between Animal Measurements and Bacterial Biomarkers

The Spearman correlation between the relative abundances of the differentially abundant bacterial taxa and animal measurements demonstrated that 11 taxa exhibited significantly strong correlation coefficients with at least one measurement (Figure 2). Furthermore, three bacterial taxa within a single taxonomic lineage (Bacteroidales F082, Limimorpha, and Limimorpha sp900318085) correlated with the backfat thickness and meat yield index in contrasting ways (r ≤ −0.8 [backfat thickness] and r ≥ 0.8 [meat yield index]; *p* ≤ 0.05). Another taxonomy lineage within the Planctomycetota phylum, including Thermoguttaceae and Thermoguttaceae DSXL01, showed a negative correlation with the backfat thickness.

## 4. Discussion

Forage feeding during the late fattening period enhances the productivity and overall health of beef cattle. A previous study has indicated that forage comprises a substantial portion of the cattle diet, ranging from 40% to 100%, and is critical for maintaining the health and performance of animals [38]. Moreover, mixing forage and concentrate to meet the nutritional requirements of beef cattle during the fattening period is important [39]. The long-term feeding of high-concentrate diets can affect the ruminal pH and bacterial communities, underscoring the importance of including forage in diets for optimal rumen health [40].

Previous studies have revealed that the ruminal microbiota composition can influence the carcass quality, meat attributes, and feed efficiency in beef cattle [41,42]. Differences in the ruminal microbiome have been linked to variations in feed efficiency, suggesting that modifying the rumen microbial functions could enhance nutrient utilization and improve the feed efficiency [41]. Sequencing techniques have been utilized to explore the correlation between the ruminal microbiota and carcass characteristics of beef cattle [42]. Dietary interventions such as feeding whole-plant corn silage improve ruminal fermentation and production performance in beef cattle [43].

The ruminal microbiome plays a significant role in the feed efficiency by converting nutrients into energy, thereby influencing meat production [44]. Alpha-diversity indices of rumen bacterial and archaeal communities have been associated with variations in feed efficiency, with more complex and diverse microbial communities observed in less efficient individuals [45]. This could explain the numerically lowest average daily gain of IRS-fed steers in this study. A previous study indicated that the feed efficiency in beef cattle is independent of the backfat thickness and feeding frequency [46]. The diversity of ruminal microbiota has been correlated with meat quality traits such as the marbling score, with animals producing highly marbled meat exhibiting a more diverse prokaryotic microbiota in their rumen [11]. However, the relationship between alpha-diversity indices and carcass characteristics, such as the meat yield and quality, needs further investigation.

Although not extensively studied in the ruminal ecosystem, the phylum Planctomycetota digest complex carbon sources [47], which is consistent with the findings from a metagenomic analysis of the camel rumen [48]. This phylum may influence the ruminal microenvironment by producing small bioactive molecules such as stieleriacines [49]. Thermoguttaceae, a family within Planctomycetota, contains nitrogenase genes [50], potentially contributing to additional nitrogen fixation in the rumen. Theoretically, the ammonia produced by nitrogenase activity can be utilized by ruminal microbes to synthesize amino acids and proteins, thereby benefiting ruminants. However, their prevalence was higher in the ruminal bacteriota of IRS-fed steers with a high crude protein content, suggesting that the additional contribution of this nitrogen-fixing bacterial population may be limited, as indicated by Postgate et al. [51] in sheep.

The genus *Limimorpha*, which was differentially abundant in IRS-fed Hanwoo steers in this study, and its species have been identified in the chicken gut [52] and equine feces [53], but their metabolic contributions to the ruminal ecosystem have not been extensively discussed. A previous study highlighted that *Limimorpha* has the most viral defense systems among metagenome-assembled genomes [54]. Despite the lack of detailed metabolic function data, our previous in vitro experiments using the same forage sources exhibited that IRS-fed batch cultures produced the most ammonia nitrogen because of their high crude protein content. This may be associated with a rise in prophages in the rumen [55] and with an increase in virus-resistant bacteria such as *Limimorpha* in the ruminal bacteriota of IRS-fed animals. Future studies should evaluate this taxonomic lineage as a potential biomarker for increasing the meat yield index.

The uncultured bacterial species *W. bifida*, which is dominant among ruminal bacteriota of RYE-fed Hanwoo steers, produces butyrate and medium-chain fatty acids by degrading xylose [56]. Increasing the levels of ruminal medium-chain fatty acids (particularly C10:0 and C12:0) can reduce protozoan numbers and ruminal ammonia utilization in cattle fed with high-grain diets in vitro [57]. Although studies on the effect of medium-chain fatty acids produced by ruminal bacteria on the meat yield in beef cattle are limited, existing evidence suggests the potential benefits of altering the fatty acid composition and modulating the ruminal microbiome. The dominance of *Succinivibrio* and its species *S. dextrinosolvens* in the ruminal bacteriota of IRS-fed cattle, accompanied with their significant correlation to backfat thickness, could be attributed to the low fiber content in the forage, providing a suitable niche for these species to access the starch content, consistent with the findings from a previous study [58]. Despite being inconsistent with the findings of this study, a previous study observed a significant positive correlation between *Succinivibrio* and the high intramuscular fat and fatty acid composition in lambs [59].

The predicted bacterial functions indicated that the differential abundance of antioxidant enzymes, including superoxide dismutase, could be an additional benefit of IRS feeding, as enhanced antioxidative function in beef cattle may be associated with better health and performance. Grass-fed beef cattle have higher superoxide dismutase activity than grain-fed beef cattle, suggesting potential differences in the antioxidant content based on the diet [60]. Furthermore, bacterial vitamin B2 synthesis mediated by the BAR-dominant flavin biosynthesis I pathway was partially supported by the increased yield grade in finishing steers fed with vitamin B2 supplements [61], although only a numerical difference was observed in this study.

## 5. Conclusions

Feeding Hanwoo steers with different forages during the late fattening stage did not significantly alter the overall bacteriome, but affected the alpha-diversity indices and specific bacteriome-derived biomarkers, which may affect the backfat thickness and the corresponding meat yield index in Hanwoo steers. The differential biomarkers identified in the ruminal bacteriome could be correlated with the feed efficiency and ruminal fermentation. Therefore, it is essential to choose appropriate forage sources during late fattening stages, considering ruminal bacteriome shifts, host genetics, diet, and management practices to optimize Hanwoo beef cattle production and enhance carcass characteristics.

## Figures and Tables

**Figure 1 microorganisms-12-02082-f001:**
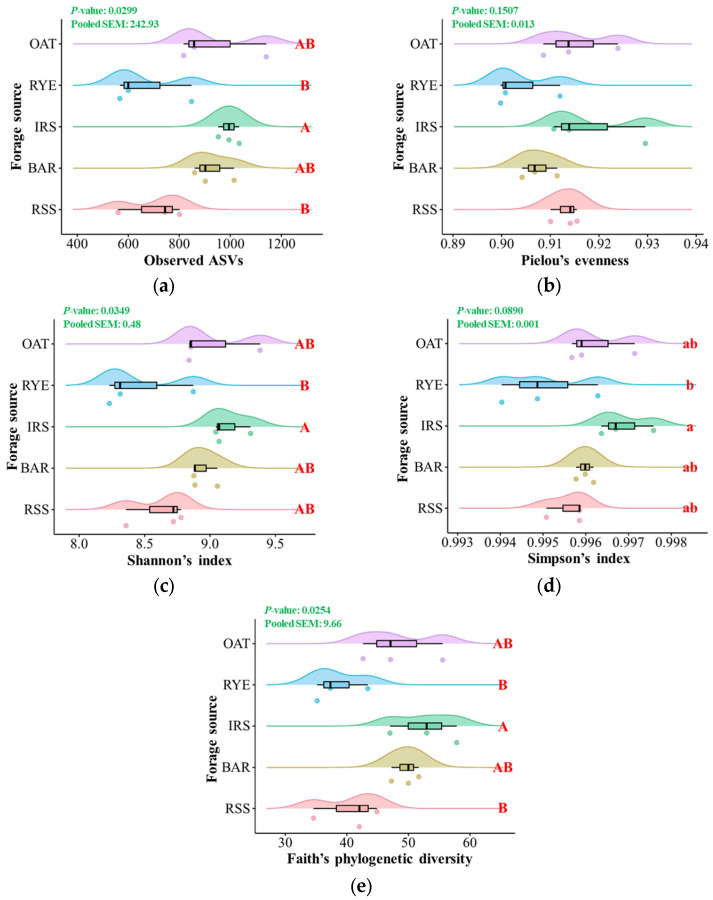
Effect of forage sources on the alpha-diversity measurements from the ruminal bacteriota of Hanwoo steers. A, B, the significant differences (*p* ≤ 0.05) among the forage sources. a, b, the tendencies (0.05 < *p* ≤ 0.10) among the forage sources. OAT, oat hay; RYE, rye silage; IRS, Italian ryegrass silage; BAR, barley forage; RSS, rice straw silage. (**a**) observed ASVs, (**b**) Pielou’s evenness, (**c**) Shannon’s index, (**d**) Simpson’s index, and (**e**) Faith’s phylogenetic diversity.

**Figure 2 microorganisms-12-02082-f002:**
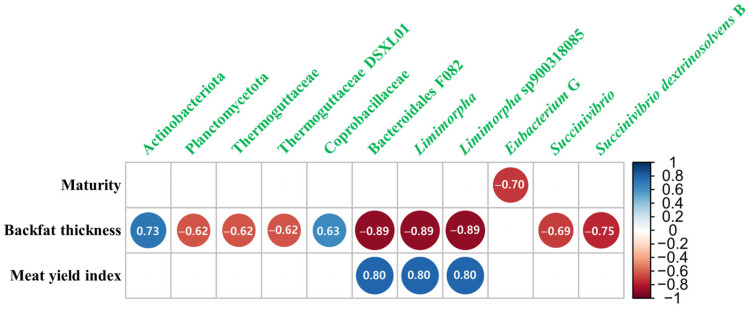
Visualized correlations between carcass traits and major bacterial taxa at the phylum, family, genus, and species levels. Only significant correlations (|r| ≥ 0.6, *p* ≤ 0.05) based on the Spearman correlation coefficients are visualized.

**Table 1 microorganisms-12-02082-t001:** Chemical composition of the concentrate and forages used for feeding the Hanwoo steers.

Items	Concentrate	Forage Sources				
		Oat Hay	Rye Silage	Italian Ryegrass Silage	Barley Forage	Rice Straw Silage
Dry matter, %	89.1	90.6	80.9	76.8	59.9	80.3
Moisture, %	10.9	9.4	19.1	23.2	40.1	19.7
Crude protein, %	14.6	6.3	8.1	10.1	6.5	3.7
Ether extract, %	4.2	1.5	3.4	2.7	1.1	1.8
Crude ash, %	6.3	6.2	9.1	19.1	9.0	17.8
Crude fiber, %	6.1	21.2	22.0	16.2	15.1	23.0
Neutral detergent fiber, %	28.6	50.6	49.1	36.2	43.1	47.2
Acid detergent fiber, %	10.7	31.6	27.9	20.1	28.2	26.7
Non-fiber carbohydrate, %	35.5	26.0	11.2	8.8	0.2	9.8
Nitrogen-free extract, %	58.0	55.4	38.4	28.8	28.3	34.0

**Table 2 microorganisms-12-02082-t002:** Effect of the forage source on the live body weight, average daily gain, and carcass characteristics of Hanwoo steers.

Measurements	Forage Sources					Pooled SEM	*p*-Value
	OAT	RYE	IRS	BAR	RSS		
Live body weight, kg							
23 months	667.33	683.00	667.67	685.67	692.67	28.91	0.9537
24 months	687.00	702.00	680.33	704.33	711.00	30.87	0.9482
25 months	708.00	735.33	695.00	727.67	736.67	32.85	0.8647
26 months	730.00	753.00	716.33	751.67	751.33	34.67	0.9169
27 months	748.00	768.67	734.67	765.67	773.67	35.35	0.9202
28 months	783.00	794.33	761.00	790.67	808.67	35.76	0.8932
Average daily gain, kg	0.771	0.742	0.622	0.700	0.773	0.057	0.2936
Intramuscular fat	8.67	7.67	7.33	6.67	8.00	0.181	0.2845
Meat color	4.00	4.67	4.00	4.67	4.67	0.067	0.1832
Fat color	3.00	3.00	3.00	3.00	3.00	0	-
Texture	1.00	1.33	1.33	1.67	1.00	0.102	0.3480
Maturity	2.67	3.00	2.67	3.00	3.00	0.067	0.5200
Backfat thickness, mm	19.33 ^AB^	22.67 ^A^	13.00 ^B^	16.33 ^AB^	20.33 ^AB^	1.94	0.0412
Longissimus dorsi muscle area, cm^2^	91.00	92.00	93.33	91.00	101.00	5.65	0.7006
Carcass weight, kg	465.67	496.33	462.00	471.33	493.33	22.73	0.6255
Meat yield index	62.75 ^ab^	61.65 ^b^	64.60 ^a^	63.47 ^ab^	62.89 ^ab^	0.666	0.0992
Meat yield grade *	C (3)	C (3)	B (2), C (1)	B (2), C (1)	C (3)	-	-
Meat quality grade ^#^	1++ (3)	1++ (2), 1+ (1)	1++ (2), 1+ (1)	1++ (1), 1+ (2)	1++ (3)	-	-
Price, KRW	20,388	17,936	20,336	17,707	18,484	1273.2	0.4313

* B and C represent the meat yield grade [15], and the number of steers associated with each meat yield grade are indicated in parentheses. ^#^ 1++ and 1+ represent the meat quality grade [15], with the number of steers indicated in parentheses. A, B, the significant differences (*p* ≤ 0.05) among the forage sources. a, b, the tendencies (0.05 < *p* ≤ 0.10) among the forage sources.

**Table 3 microorganisms-12-02082-t003:** Effect of forage sources on the overall bacterial communities and their corresponding functional profiles analyzed using permutational multivariate analysis of variance test.

Input Normalized Abundance Profile	Distance Matrix	Permutations	R^2^ of Forage Sources	Pseudo-F	*p*-Value
Average rarefied ASV	Bray–Curtis	9999	0.2617	0.8864	0.6381
	Jaccard	9999	0.2868	1.0052	0.4421
KEGG orthologs	Bray–Curtis	9999	0.3259	1.2085	0.3240
	Jaccard	9999	0.2109	0.6682	0.8906
Enzyme commissions	Bray–Curtis	9999	0.2979	1.0608	0.4079
	Jaccard	9999	0.2328	0.7587	0.8176

There were no significant pairwise comparisons for all abundance profiles based on the Benjamini–Hochberg-corrected *p*-value (q > 0.10). ASV, amplicon sequence variance.

**Table 4 microorganisms-12-02082-t004:** Dominant bacterial taxa considering the feeding of different forage sources.

Bacterial Taxon		Forage Sources							
		Relative Abundance (%)							
**Phylum**	**Dominance**	**OAT**	**RYE**	**IRS**	**BAR**	**RSS**	**Pooled SEM**	**LDA score**	***p*-value**
Actinobacteriota	RSS	0.407	0.675	0.300	0.207	** 0.794 **	0.306	3.612	0.0273
Planctomycetota	IRS	0.288	0.060	** 0.405 **	0.206	0.147	0.189	3.431	0.0374
**Family**	**Dominance**	**OAT**	**RYE**	**IRS**	**BAR**	**RSS**	**Pooled SEM**	**LDA score**	***p*-value**
Coprobacillaceae	RSS	0.142	0.450	0.065	0	** 0.724 **	0.645	3.556	0.0497
Thermoguttaceae	IRS	0.269	0.060	** 0.391 **	0.206	0.147	0.170	3.367	0.0374
Dethiosulfovibrionaceae	RSS	0	0.139	0	0.051	** 0.172 **	0.162	3.098	0.0479
**Genus**	**Dominance**	**OAT**	**RYE**	**IRS**	**BAR**	**RSS**	**Pooled SEM**	**LDA score**	***p*-value**
*Weimeria*	RYE	0.001	** 0.535 **	0.134	0	0.201	0.568	3.434	0.0263
*Succinivibrio*	IRS	0.846	0.238	** 1.433 **	0.808	0.070	0.642	3.842	0.0277
*Limimorpha*	IRS	0.268	0.085	** 1.206 **	0.741	0.167	0.996	3.756	0.0377
*Shuttleworthia*	IRS	0.075	0	** 0.285 **	0.049	0	0.394	3.197	0.0423
*Eubacterium* G	OAT	** 0.112 **	0	0.037	0	0	0.141	2.819	0.0262
**Species**	**Dominance**	**OAT**	**RYE**	**IRS**	**BAR**	**RSS**	**Pooled SEM**	**LDA score**	***p*-value**
*Succinivibrio dextrinosolvens* B	IRS	0.727	0.080	** 1.252 **	0.718	0.070	0.630	3.765	0.0262
*Weimeria bifida*	RYE	0	** 0.421 **	0.062	0	0.185	0.429	3.339	0.0331

Only completely classified taxa are presented. The dominance of each taxon for specific forage feeding is additionally shown as bolded and underlined numbers. SEM—standard error of means; LDA score—linear discriminant analysis. OAT, oat hay; RYE, rye silage; IRS, Italian ryegrass silage; BAR, barley forage; RSS, rice straw silage.

**Table 5 microorganisms-12-02082-t005:** Dominant predicted enzyme commissions and MetaCyc pathways considering the feeding of different forage sources.

EC Number		Forage Sources								
		Relative Abundance (%)								
	Dominance	OAT	RYE	IRS	BAR	RSS	Pooled SEM	LDA Score	*p*-Value	Description
EC:1.1.1.22	BAR	0.127	0.125	0.132	** 0.134 **	0.115	0.014	2.121	0.0477	UDP-glucose 6-dehydrogenase
EC:1.15.1.1	IRS	0.148	0.130	** 0.151 **	0.147	0.132	0.022	2.227	0.0382	Superoxide dismutase
EC:2.1.1.170	RYE	0.160	** 0.162 **	0.159	0.157	0.162	0.003	2.187	0.0285	16S rRNA (guanine(527)-N(7))-methyltransferase
EC:2.3.1.30	RYE	0.165	** 0.175 **	0.163	0.163	0.172	0.013	2.101	0.0200	Serine O-acetyltransferase
EC:2.4.2.10	RYE	0.198	** 0.201 **	0.195	0.192	0.200	0.007	2.168	0.0421	Orotate phosphoribosyltransferase
EC:2.7.1.71	RSS	0.197	0.202	0.194	0.192	** 0.203 **	0.008	2.002	0.0433	Shikimate kinase
EC:2.7.7.60	RYE	0.215	** 0.227 **	0.208	0.205	0.226	0.017	2.232	0.0310	2-C-methyl-D-erythritol 4-phosphate cytidylyltransferase
EC:3.4.11.4	RSS	0.155	0.160	0.151	0.149	** 0.165 **	0.010	2.084	0.0302	Tripeptide aminopeptidase
EC:3.5.99.10	RSS	0.165	0.170	0.162	0.165	** 0.171 **	0.005	2.020	0.0203	2-iminobutanoate/2-iminopropanoate deaminase
EC:3.6.1.23	RYE	0.159	** 0.162 **	0.158	0.157	0.161	0.002	2.014	0.0192	dUTP diphosphatase
EC:4.1.1.20	RYE	0.175	** 0.181 **	0.171	0.171	0.178	0.007	2.085	0.0377	Diaminopimelate decarboxylase
EC:5.1.1.3	RYE	0.154	** 0.159 **	0.153	0.152	0.158	0.005	2.147	0.0388	Glutamate racemase
EC:5.1.1.7	BAR	0.149	0.140	0.148	** 0.151 **	0.143	0.008	2.080	0.0484	Diaminopimelate epimerase
**MetaCyc** **pathway**	**Dominance**	**OAT**	**RYE**	**IRS**	**BAR**	**RSS**	**Pooled SEM**	**LDA score**	***p*-value**	**Description**
RIBOSYN2-PWY	BAR	0.694	0.731	0.712	** 0.742 **	0.728	0.029	2.743	0.0491	Flavin biosynthesis I (bacteria and plants)

Dominance of each functional feature for specific forage feeding is presented as bolded and underlined numbers. OAT, oat hay; RYE, rye silage; IRS, Italian ryegrass silage; BAR, barley forage; RSS, rice straw silage.

## Data Availability

All raw 16S rRNA gene sequencing data can be availed by the authors upon request.

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
