# Peer review of "Impact of Forage Sources on Ruminal Bacteriome and Carcass Traits in Hanwoo Steers During the Late Fattening Stages"

_microorganisms, 2024, doi:10.3390/microorganisms12102082_

Round 1
Reviewer 1 Report
Comments and Suggestions for Authors
This study examined the effects of different forage sources on the ruminal bacteriome, growth performance, and carcass characteristics of Hanwoo steers during the fattening stage. It is essential to choose appropriate forage sources during late fattening stages, considering ruminal bacteriome shifts, host genetics, diet, and management practices to optimize Hanwoo beef cattle production and enhance carcass characteristics.
However, there are some issues that need to be revised and improved. Specifically, the following are the review comments:
1.The titles in Table 1 are not aligned with the data below. Please pay attention to the format and layout of the table.
2.The animals were fed with OAT, RYE, IRS, BAR, and RSS at 1.5 kg dry matter and 9.0 kg dry matter of concentrate per day. Can the experiment ensure that all of these are consumed? Is there any record of actual feed intake and remaining feed intake? If there is no record, different dry matter intake consumed may have an impact on daily weight gain.
3.The Hanwoo cows in the caption of Figure 1 are not accurate. The rest of the article mentions hanwoo beef cattle. But cow generally refers to the dairy cow.
4.In the discussion section, while citing other literature, it is important to compare and analyze one's own results with those of other studies.
5.The proportion of references in the past three to five years is not high, it is recommended citing more of the latest literature.
Comments on the Quality of English LanguageThe same meaning can be replaced with different words. And advanced sentence structures can be appropriately applied.
Author Response
Comments 1: This study examined the effects of different forage sources on the ruminal bacteriome, growth performance, and carcass characteristics of Hanwoo steers during the fattening stage. It is essential to choose appropriate forage sources during late fattening stages, considering ruminal bacteriome shifts, host genetics, diet, and management practices to optimize Hanwoo beef cattle production and enhance carcass characteristics.
However, there are some issues that need to be revised and improved. Specifically, the following are the review comments:
Response 1: Thank you for your interests in our manuscript. We are trying our best to revise it based on your valuable comments.
Comments 2: The titles in Table 1 are not aligned with the data below. Please pay attention to the format and layout of the table.
Response 2: Thank you for your comment. Now the Table 1 has been reformatted and the contents are aligned to their corresponded data.
Comments 3: The animals were fed with OAT, RYE, IRS, BAR, and RSS at 1.5 kg dry matter and 9.0 kg dry matter of concentrate per day. Can the experiment ensure that all of these are consumed? Is there any record of actual feed intake and remaining feed intake? If there is no record, different dry matter intake consumed may have an impact on daily weight gain.
Response 3: Thank you for your comment. In this study, we attempted to calculate intake by measuring the feed supply and residual amount, but since restricted feeding was used, there was no residual amount during the experimental period. Therefore, it is thought that there will be no problem in presenting dry matter intake as 9 kg for compound feed and 1.5 kg for forage, and this will not have any effect on the average daily gain. Additionally, we will add a phrase about dry matter intake measurement (Dry matter intake (DMI) was measured for three consecutive days each month during the experiment, and the DMI was determined by measuring the remaining feed before the morning feeding) to the Materials and Methods (Line 84 - 86).
Comments 4: The Hanwoo cows in the caption of Figure 1 are not accurate. The rest of the article mentions hanwoo beef cattle. But cow generally refers to the dairy cow.
Response 4: Thank you for pointing this out. Now it is revised to ‘Hanwoo steers’ in the figure 1 legend (Line 246).
Comments 5: In the discussion section, while citing other literature, it is important to compare and analyze one's own results with those of other studies.
Response 5: Thank you for your valuable comment. We carefully reviewed the discussion section and added more detailed explanations where our results were not thoroughly described or clearly supported.
Comments 6: The proportion of references in the past three to five years is not high, it is recommended citing more of the latest literature.
Response 6: Thank you for your comment. While we found a limited number of relevant recent studies, we have added two additional references to address your suggestion (Line 31, 44).
Reviewer 2 Report
Comments and Suggestions for Authors
Line 74 Table 1 needs to be introduced in the text. Place it in section 2.1 after being introduced.
Line 74 Table 1 - If the table refers to the chemical composition of feedstuff, the DM intake should not be there. You can however adapt the caption to include this content.
Line 74 Table 1 please include moisture content
Line 74 Table 1 While it is important that all acronyms in a table are explained, you don’t need to use acronyms if you are spelling out the words.
Line 82 These acronyms need to be spelled out the first time used in the text. Please mind you have done that in the table however tables, figures and text should stand alone from each other with this regard. I suggest spelling out in the table and spelling out again in the text followed by the acronym within bracket. If you want to use the acronyms in a table you have to spell them out in notes in the bottom of the table.
Line 88 “preheated thermal bottle” Please give temperature.
Line 107 Please explain how you have calculated the meat yield index
Line 112 if the quality grades were based on marbling score only, why grading?,… doesn’t the marbling score give the grade already? Please clarify.
Line 129 Please explain how these alpha-diversity measurements are calculated or give a reference.
Line 144 Please replace “equality” of variances by “homogeneity” of variances. Adjust on the text throughout.
Line 146 Please specify what the GLIMMIX procedure does. I believe it is an ANOVA therefore you also need to state what post hoc test was used. Also, if you have tested the normal distribution of variables as a pre-requisite of the ANOVA this is wrong. What you should have done was checking the normal distribution of the residuals. I have to alt this review at this point. Please review your methods and adjust accordingly before resubmitting.
Line 161 why using Perason’s correlation? are all the variables normally distributed?
Author Response
Comments 1: Line 74 Table 1 needs to be introduced in the text. Place it in section 2.1 after being introduced.
Response 1: Thank you for your comment. Now Table 1 is placed after mentioning in section 2.1. Not to make truncation of the content, Table 1 is placed at the beginning of next page.
Comments 2: Line 74 Table 1 – If the table refers to the chemical composition of feedstuff, the DM intake should not be there. You can however adapt the caption to include this content.
Response 2: Thank you for pointing this out. DM intake was described in the section 2.1. so we have removed the DM intake from Table 1.
Comments 3: Line 74 Table 1 please include moisture content
Response 3: Thank you for your comment. Now we have added moisture content in Table 1.
Comments 4: Line 74 Table 1 While it is important that all acronyms in a table are explained, you don‘t need to use acronyms if you are spelling out the words.
Response 4: Thank you for pointing this out. Now all acronyms were removed from Table 1.
Comments 5: Line 82 These acronyms need to be spelled out the first time used in the text. Please mind you have done that in the table however tables, figures and text should stand alone from each other with this regard. I suggest spelling out in the table and spelling out again in the text followed by the acronym within bracket. If you want to use the acronyms in a table you have to spell them out in notes in the bottom of the table.
Response 5: Thank you for pointing this out. We have mentioned the full name in the introduction part which we firstly introduced the abbreviations of each forage type. To make sure, we have added the full name once again in the first appearance of those in the Materials and Methods section (Line 81 - 82).
Comments 6: Line 88 “preheated thermal bottle” Please give temperature.
Response 6: Thank you for your comment. Now we have added preheat temperature (39°C) in Line 90.
Comments 7: Line 107 Please explain how you have calculated the meat yield index
Response 7: Thank you for your comment. Now we have added an equation to calculate meat yield index based on the carcass measurement with corresponded reference (Line 114 - 115).
Comments 8: Line 112 if the quality grades were based on marbling score only, why grading?,… doesn‘t the marbling score give the grade already? Please clarify.
Response 8: Thank you for your comment. In Korea, it is more common to report the meat quality grade rather than the marbling score. The marbling score is used to calculate a simple quality grade, ranging from 2 to 1++, which is a categorical variable that consumers are more familiar with. While scientifically, providing just the marbling score would suffice, reporting the quality grade is a widely accepted practice in the Korean market.
Comments 9: Line 129 Please explain how these alpha-diversity measurements are calculated or give a reference.
Response 9: Thank you for your comment. Now we have added all the references for all the indices we used (Line 139 - 140).
Comments 10: Line 144 Please replace “equality” of variances by “homogeneity” of variances. Adjust on the text throughout.
Response 10: Thank you for your comment. We have fixed the term as the reviewer suggested (Line 153).
Comments 11: Line 146 Please specify what the GLIMMIX procedure does. I believe it is an ANOVA therefore you also need to state what post hoc test was used. Also, if you have tested the normal distribution of variables as a pre-requisite of the ANOVA this is wrong. What you should have done was checking the normal distribution of the residuals. I have to alt this review at this point. Please review your methods and adjust accordingly before resubmitting.
Response 11: Thank you for your valuable comment. We have double-checked all the normality test was done from residuals. P-values of forage source effect for intramuscular fat and texture was now revised (Table 2). And we have also revised the post hoc test we used (Line 156-157).
As the reviewer asked, GLIMMIX is usually used to include random effect and can include error terms that are not normally distributed. Since we separated the statistical analysis depending on the distribution of residuals now, we have revised the statistical part (Line 155). The overall statistical conclusions were not different between GLIMMIX and one-way ANOVA for normally distributed data.
Comments 12: Line 161 why using Perason‘s correlation? Are all the variables normally distributed?
Response 12: Thank you for your comment. We have double-checked the distribution of the data used in the correlation analysis and re-analyzed the correlation plot using Spearman’s correlation coefficients (Line 171, 232, 235). Statistical analysis part is now revised as the reviewer suggested (Line 171) with corresponded changes in result part (Line 236 - 243). You can find revised plots in Figure 2.
Round 2
Reviewer 2 Report
Comments and Suggestions for Authors
Dear authors,
I urge you to seek advice from a statistician. The issues I have raised before were partially addressed, raising concerns. Please address the statistical issues raised. Check throughout the paper. I can not progress further with the revision at this moment.
Author Response
Dear Reviewer,
Thank you for your valuable feedback. We have taken careful consideration of your comments and sought advice from a statistician with extensive experience in in vivo experiments. As a result, we have implemented the following revisions to address the statistical concerns raised (We have highlighted the revised sections shaded in yellow):
Normality Assessment: We conducted a comprehensive normality assessment for all variables using the Shapiro-Wilk test and visual inspection of histograms (PROC UNIVARIATE). This informed our decision to utilize either parametric or non-parametric analyses.
Mixed Model Analysis: For variables meeting normality assumptions, we employed a mixed model approach (PROC MIXED) with Treatment as a fixed effect and Replication as a random effect. Degrees of freedom were calculated using the Kenward-Roger method. The model was specified as follows:
Yij=μ+Ti+Rj+ϵij
Where Yij is the response variable, μ is the overall mean, Ti is the fixed effect of the ith treatment, Rj is the random effect of the jth replication, and εij is the random error.
Non-Parametric Analysis: For variables that did not meet normality assumptions, we performed a non-parametric analysis using the Kruskal-Wallis test (PROC NPAR1WAY with the WILCOXON option). Exact p-values were computed to compare these variables across treatments.
Pooled SEM Issue: We identified and corrected an issue related to the Pooled SEM based on the statistician’s suggestions and the SAS results. This modification ensures that our analysis adheres to standard procedures and accurately reflects the data.
We are confident that these revisions address the statistical concerns raised and enhance the robustness of our analysis. Your guidance is greatly appreciated, and we eagerly await your further feedback.